# Blood Flow in the Internal Jugular Veins in the Lateral Decubitus Body Position in the Healthy People

**DOI:** 10.3390/jcm14041211

**Published:** 2025-02-12

**Authors:** Agata Maria Kawalec-Rutkowska, Joanna Czaja, Marcin Skuła, Marian Simka

**Affiliations:** Institute of Medical Sciences, University of Opole, 45-060 Opole, Poland; agata.kawalec@uni.opole.pl (A.M.K.-R.); heyna@uni.opole.pl (J.C.); mskula@uni.opole.pl (M.S.)

**Keywords:** internal jugular vein, blood flow, body position

## Abstract

**Background:** Some studies have suggested that the lateral decubitus position during sleep may protect the brain from neurodegenerative processes. Although the mechanisms of such possible protection are not known, an optimal venous outflow may be responsible. Venous outflow from the cranial cavity is dependent on the body’s position. However, to date, flow in the internal jugular veins (IJVs) in the lateral position has not been studied quantitatively. **Methods:** Using ultrasonography, we measured the cross-sectional areas and flow volumes in the IJVs in a group of 25 healthy individuals aged 20–52 ± 12.1 years. These measurements were performed in the supine, upright, and lateral decubitus positions. **Results:** In the lateral decubitus position, we revealed a collapse of the IJV located higher, dilatation of the opposite vein, and a shift in flow from one vein to the opposite. In the right lateral position, the mean cross-sectional area and flow in the right IJV were 88.6 ± 71.1 mm^2^ and 74.3 ± 97.5 mL/min, in the left IJV: 37.2 ± 33.4 mm^2^ and 48.8 ± 82.8 mL/min. In the left lateral position, the right IJV was 38.4 ± 30.7 mm^2^ and 56.7 ± 56.1 mL/min, and the left IJV was 75.9 ± 51.9 mm^2^ and 99.7 ± 123.9 mL/min. However, there was also a high heterogeneity of the cross-sectional area changes, and in many participants, this pattern was not observed. Regarding flow volumes in the lateral body positions, in comparison with the supine position, the total outflow through both internal jugular veins was not significantly different. **Conclusions:** In terms of venous outflow, the lateral decubitus position did not differ significantly from the supine position. The working hypothesis of a potentially protective effect of this body position during sleep against neurodegeneration through improved venous outflow has not been proven, at least in healthy individuals.

## 1. Introduction

Currently, there is a growing body of evidence that neurodegenerative diseases are associated with impaired functioning of the glymphatic system in many patients. Less efficient cleansing of the cerebral parenchyma from pathological or misfolded proteins, such as α-synuclein, β-synuclein, β-amyloid, or bacterial curli, is thought to be one of the important triggers of the neurodegenerative process [1,2,3,4,5]. Since the glymphatic system is primarily active during sleep, optimal conditions for this system are important [6,7,8,9,10,11]. Of note, some studies have suggested that body position during sleep may have an impact on the development of neurodegeneration [12,13,14]. Interestingly, humans prefer to sleep in either the supine or lateral position [15,16], and the lateral position seems to exhibit a protective effect. Possible mechanisms responsible for such a property of the lateral body position, for the time being, remain elusive. Nonetheless, our team has already proposed that perhaps impaired cerebral venous drainage can provide a reasonable explanation [17].

Importantly, cerebral veins, as well as extracranial venous outflow routes, including the internal jugular veins, are closely linked to glymphatic system function since they provide key drainage pathways for metabolic waste removal from the brain parenchyma. During activation of the glymphatic system, cerebrospinal fluid enters the parenchyma through the periarterial spaces. This process is facilitated by aquaporin-4 water channels located on astrocytes. Then, the cerebrospinal fluid mixes with the interstitial fluid, and this process enables the removal of the above-mentioned toxic proteins. Since the cerebrospinal fluid-interstitial fluid mixture moves along the perivenous spaces, any disruption to the venous outflow may potentially contribute to glymphatic dysfunction and an increased risk of neurodegenerative disease [18,19]. It has already been demonstrated that activation of the astroglial-mediated interstitial fluid bulk flow is initiated by a temporary decrease in the cortical blood flow (mediated by the noradrenergic neurons of the locus ceruleus in the brain stem). This activation, in turn, facilitates the inflow of cerebrospinal fluid into the cranial cavity from the vertebral canal and then into the perivascular space, which initiates the function of the glymphatic system [9,20,21,22]. According to the Kellie-Monroe doctrine, which states that the total volume inside the cranial cavity is constant, an unrestrained influx of cerebrospinal fluid to the cranial cavity would require that a similar volume of venous blood flows out from the cranial cavity. Any obstacle compromising venous outflow, at least theoretically, may impair the glymphatic system. Indeed, abnormal venous outflow through the internal jugular veins (IJVs), which are the main blood vessels draining the brain, has been revealed in many neurodegenerative and neuroinflammatory diseases such as multiple sclerosis, Parkinson’s disease, lateral amyotrophic sclerosis, and Ménière disease [23,24,25,26,27,28,29,30,31]. Yet, actually, there are two alternative outflow routes from the cranial cavity: one comprising the IJVs and the second one built up by the vertebral veins, vertebral epidural plexus, and other deep cervical veins. In the supine body position, the majority of blood flows out through the IJVs. In the upright body position, because of gravitational effects, these veins collapse and the outflow is shifted toward the vertebral and deep cervical veins, which are characterized by a higher flow resistance but do not collapse easily when the head is elevated above the heart level [20,21,22,23,24,25,26,27,28,29,30,31,32,33,34,35,36]. Interestingly, outflow through the IJVs has not been thoroughly studied in the lateral decubitus position.

In our previous pilot study [17], we found that in the lateral decubitus position, the IJV, which in this body posture was above the level of the right atrium, was partially collapsed, while the opposite one remained open and was slightly wider in comparison with the cross-sectional area of this vein in the supine body position. When the patient changed body position to the opposite lateral position, for example, from the left to the right side, the IJVs behaved following this pattern: the vein located above the heart level collapsed, and that located below it dilatated. However, in this preliminary study, the IJV was not examined quantitatively, and only a few individuals were assessed. The current study was aimed at quantitative evaluation of the flow and diameters in the IJVs in different body positions, particularly in the lateral decubitus positions, in order to better understand flow phenomena in these veins occurring when people are lying down on their side.

## 2. Materials and Methods

This study was an observational research performed on healthy subjects. For this purpose, we recruited 25 healthy individuals, 21 women and 4 men. They were recruited from among the employees of our university. They were aged 20–52 years (mean 33.5 years). None of them presented with significant co-morbidities, including obesity.

The inclusion criteria of this study were as follows:participant’s age > 18 years,no co-morbidities.

The exclusion criteria were as follows:
diagnosis of neurodegenerative disease, such as Parkinson’s disease, Alzheimer’s disease, or lateral amyotrophic sclerosis,history of cerebral stroke,history of surgical or endovascular treatment of carotid or vertebral arteries,history of cerebral disease of an inflammatory, infectious, or vascular etiology,clinically relevant circulatory or respiratory insufficiency.

In all of them, ultrasonographic examinations of the IJVs in the middle portion of these veins (at the level of the thyroid cartilage) were performed. All measurements were performed at the same level, which was marked on the skin with a white crayon. These examinations were performed on a flat examination table, and during measurements, the participants were asked to breathe normally. Measurements of the cross-sectional areas and flow volumes were performed on both sides of the neck. The cross-sectional areas were calculated in a standard manner after freezing the picture of the target vein and using the measuring application of the ultrasound system. The mean velocity flow was evaluated during one respiratory cycle, and the flow volume was calculated using the values of the cross-sectional area and mean velocity flow. During the scanning, participants were asked not to rotate their heads in the lateral body position. All the above-described measurements were performed in four body positions:supine,sitting,right lateral decubitus, i.e., lying on the right side,left lateral decubitus, i.e., lying on the left side.

We also checked the internal jugular veins to see if there were any anatomical abnormalities, such as stenoses, aneurysmatic dilatations, external compressions, pathologic jugular valves, or other intraluminal pathologies. Considering the difficulties associated with reliable measurement of the flow in the vertebral and deep cervical veins, we did not evaluate the cross-sectional areas and flow velocities in these veins, constituting an alternative outflow route. Instead, we exclusively focused on the total flow in the IJVs. All examinations were performed with the GE Versana Active ultrasound system (GE HealthCare, Chicago, IL, USA), using the 10 MHz linear probe and venous preset of the ultrasound system.

Informed consent was obtained from all individuals participating in this survey. The study was conducted in accordance with the Declaration of Helsinki and approved by the Bioethical Committee of the University of Opole; approval No UO/0012/KB/2022.

### Statistical Analysis

In order to statistically assess the differences between cross-sectional areas and flow volumes in the examined veins, the paired *t*-test and the one-way ANOVA test were used. We also calculated the statistical significance of the differences in means in order to assess the clinical importance of the revealed divergences in cross-sectional areas and flow volumes. The normality of the data was assessed using the Shapiro−Wilk test, which revealed that the distribution of the samples was not significantly different from a normal distribution. The significance of *p* values for all statistical tests used was set at *p* < 0.05. Statistical analyses were performed using the PAST data analysis package (version 3.0; University of Oslo, Norway).

## 3. Results

We did not find structural abnormalities, such as aberrant valves or other pathological intraluminal structures, in any of the IJVs assessed. No thrombotic occlusions were observed in any of the studied veins.

In general, the IJV followed the already revealed pattern of partial or complete collapse in the sitting body position and dilatation in the supine and lateral decubitus positions. In the supine body position, the mean cross-sectional area of the right IJV was 96.2 ± 54.4 mm^2^, and that of the left IJV was 64.8 ± 39.6 mm^2^. These differences were statistically significant (paired *t*-test: *p* = 0.03; test for difference in means: *p* = 0.02). This finding is not new, and it is well known that the right IJV is usually wider, which is associated with the embryological development of these veins [35]. In the sitting body position, the mean cross-sectional area of the right IJV was 10.8 ± 6.1 mm^2^, and of the left IJV was 10.3 ± 9.3 mm^2^. These differences were insignificant (test for difference in means: *p* > 0.05).

In the right lateral decubitus body position, the mean cross-sectional area of the right IJV was 88.6 ± 71.1 mm^2^, and that of the left IJV was 37.2 ± 33.4 mm^2^. In the left lateral decubitus body position, the mean cross-sectional area of the right IJV was 38.4 ± 30.7 mm^2^, and that of the left IJV was 75.9 ± 51.9 mm^2^. These differences between cross-sectional areas in the lateral body position between the right and left IJV were significant; the test for difference in means: *p* = 0.002 (the right lateral decubitus position) and *p* = 0.003 (the left one position).

Details of the statistical analysis and graphical representation are shown in Figure 1. It can be seen that, on average, the right IJVs were larger in comparison with the left ones; both IJVs almost totally collapsed in the sitting body position, and in the lateral body position, one IJV constituted about 2/3 of the total outflow area—the right IJV in the right lateral decubitus position, and the left IJV in the left lateral decubitus position. The differences between the cross-sectional areas at particular body positions were statistically significant (*p* < 0.001). In addition, the cross-sectional area of each IJV significantly differed (*p* < 0.001) between the right and left lateral decubitus positions, as demonstrated using the paired *t*-test.

We also found that the flow in the right and left IJV followed the same pattern: it was higher in the supine body position and in the lateral decubitus position ipsilateral to the vein studied. In the supine body position, the flow through the right IJV was 101.1 ± 95.7 mL/min, and through the left IJV, it was 55.1 ± 61.6 mL/min. These differences between the right and left IJV were statistically significant (test for difference in means: *p* < 0.05).

In the sitting body position, the flow through the right IJV was 13.3 ± 16.7 mL/min, and of the left IJV, it was 11.9 ± 13.9 mL/min. These differences were insignificant (the test for difference in means: *p* > 0.05).

In the right decubitus body position, the flow through the right IJV was 74.3 ± 97.5 mL/min ^2^, and of the left IJV, it was 48.8 ± 82.8 mL/min. In the left decubitus body position, the flow through the right IJV was 56.7 ± 56.1 mL/min, and of the left IJV, it was 99.7 ± 123.9 mL/min. Unlike cross-sectional areas, regarding the flow volumes, differences between the right and left IJV in the lateral body position were statistically insignificant (test for difference in means: *p* > 0.05). A graphical representation is shown in Figure 2.

It can be seen that, on average, the flow through the right IJV was higher in comparison with the left IJV in the supine and right decubitus body positions, while in the left decubitus position, the left IJV became dominant. There were statistically significant differences (*p* < 0.05) in the flow through the IJV in the sitting position in comparison with other body positions (Figure 2), yet differences in the flow through the right or left IJV in the right and left lateral decubitus positions were not statistically significant (paired *t*-test: *p* > 0.05).

However, in many veins, the pattern described above was not observed. There were some IJVs that dilatated in a specific body position, but the flow in this position decreased, or vice versa; the vein reduced its cross-sectional area, but the flow increased. A graphical representation of these phenomena is shown in Figure 3.

Then, we calculated the total flow through both IJVs. In the supine body position, it was 156.1 ± 107.1 mL/min; in the sitting body position, it was 25.5 ± 18.7 mL/min; in the right decubitus body position, it was 126.1 ± 119.3 mL/min ^2^; and in the left decubitus body position, the flow was 156.4 ± 133.4 mL/min (Figure 4). Statistical analysis showed that although there were statistically significant differences revealed by the one-way ANOVA test, actually, these differences regarded the flow in the sitting position, which was significantly lower in comparison with other body positions. Analysis with the use of the paired *t*-test did not demonstrate significant differences between the total flow volumes through both IJVs in the supine and the lateral decubitus positions (*p* > 0.05).

Since there was a high heterogeneity regarding the cross-sectional areas and flow volumes in the examined veins, we also analyzed whether asymmetry of the IJVs had an association with the parameters studied. It is well known that IJVs exhibit high anatomical variability. In most people, these veins are asymmetric. The right IJV is usually wider, although in some people, the left IJV is dominant. In addition, the inflow to these veins is typically asymmetric. A symmetric confluence of the sinuses (the connection of the superior sagittal sinus, straight sinus, and occipital sinus, which divides into two transverse sinuses) is rarely observed. In the majority of people, this anatomical structure is asymmetric. Hence, the inflow into the IJVs is not equal, and usually, the cerebral outflow is primarily directed to the right IJV [35].

We categorized our participants into two groups: people with a dominant-right-IJV and those with a dominant-left-IJV. For the purpose of this analysis, the dominance of a particular IJV was defined as the vein in which the flow in the supine body position was higher than that in the opposite position. In the studied group, there were 14 participants with a dominant-right-IJV and 11 participants with a dominant-left-IJV. The flow volumes in the dominant-right-IJV vs. dominant-left-IJV individuals are shown in Figure 5. In general, the flow volumes were higher in the group with a dominant-right-IJV, although these differences were not statistically significant. There was a statistically significant flow reduction through the IJVs in the sitting position, while the differences in flow volumes between other body positions were statistically insignificant.

Finally, we calculated the mean flow volume in the right vs. left IJV for all four body positions. The flow was slightly higher on the right side: 61.4 mL/min than on the left side: 53.9 mL/min; still, this difference, evaluated by means of the paired *t*-test, was not statistically significant (*p* > 0.05). We also determined the body position with the highest outflow. The highest flow volume in the supine body position was revealed in 11 participants; in 10 individuals, it was the highest in the left lateral body position and in 4 people in the right lateral position.

## 4. Discussion

The results of our study can be summarized as follows: Firstly, there was a significant reduction in both cross-sectional areas of the IJV and flow volumes through these veins in the sitting body position in comparison with the supine position. This is not a new finding. This is a well-known phenomenon that was described two decades ago. In the supine body position, blood flows out from the cranial cavity primarily through the IJVs, which are continuations of the cerebral venous sinuses. In this body position, at least 70% of the blood flows out through the IJVs. However, when the head is elevated, approximately 80% of the blood flows out through the paravertebral route [35,36,37,38,39]. This is possible because the jugular and vertebral outflow pathways are anatomically connected in the area of the foramen magnum [35,40]. In our group of participants, we also revealed the phenomenon described above. Of note, a reduction in flow through the IJVs in the sitting position does not mean that the total cerebral blood flow in this body position is significantly lower. This change reflects a shift of the venous outflow to alternative pathways, resulting from the collapse of the IJVs, which is due to the negative transmural pressure in these veins [33,34]. Since it is difficult to measure quantitatively the flow in the vertebral veins, spinal epidural plexus, and small deep cervical veins with the use of ultrasonography, for the purpose of our investigation, it was assumed that the total flow remained unchanged and that the “missing” flow volume in the IJVs has been shifted to the other veins. Previous research has validated this assumption not only clinically but also in computer simulations [41,42].

Secondly, we found significant changes in the cross-sectional areas of the IJVs in the lateral decubitus body positions. These changes, a collapse of the vein located above the heart level and dilatation of the opposite IJV, were partially in line with the observations from our previous small study [17]. It is worth noting that such a behavior of IJVs, although predictable, has not been quantitatively studied before. Still, in a larger cohort, not only did we reveal a high heterogeneity of the cross-sectional area changes, which was manifested through very high standard deviations of this parameter, but we also found that in the lateral decubitus body position in many participants, the IJVs did not follow the above-described pattern (see Figure 3). This suggests that the position-dependent change of the shape and flow in the IJV is more complex and is determined by more factors than gravitation alone [32,43]. Below, we offer possible explanations for this phenomenon.

Thirdly, we found that in the lateral decubitus body positions, the right and left ones, the total outflow through the IJV was not significantly different from such a flow in the supine position. Although there was a shift in the flow from one IJV to the opposite vein, depending on which side the person was lying on, the total outflow remained largely unchanged. Of note, such a shift of outflow from the right to the left IJV, or vice versa, was not found in every individual studied, as can be seen in Figure 3.

In conclusion, in terms of venous outflow from the cranial cavity, the lateral decubitus position does not differ significantly from the supine position. Therefore, it can be concluded that this study does not confirm the working hypothesis presented in our previous paper [17]. However, there are several caveats to this conclusion.

In this study, we examined veins in healthy individuals. Even if the total cerebral outflow in healthy people does not depend on a particular type of horizontal body position, as our study suggests, it does not mean that in people prone to the development of neurodegeneration, the anatomy and physiology of the IJVs are the same. Therefore, a similar investigation should be performed in patients presenting with neurodegenerative diseases. Of note, in 14 of the 25 healthy individuals studied, the highest flow was found in one of the lateral decubitus body positions. However, this finding was not statistically significant in the cohort examined.

The highly unpredictable behavior of IJVs in the lateral position, which was observed in some individuals assessed in this study, should be thoroughly explained. These changes in the flow volume and vein diameters are difficult to explain if, anatomically, the IJVs and vertebral veins constitute a simple interconnected system in which only gravitation plays a role. Most likely, the reduction in flow, combined with an increased cross-sectional area or the reversed phenomenon, resulted from an interplay between anatomical asymmetry of the cerebral sinuses [35], compression of the IJV by bony structures in its upper part [44], especially while lying on one side with the head slightly distorted, and possibly with flow resistance evoked by an abnormal flow pattern in the IJV on otherwise normal jugular valves [45]. In addition, a slight head rotation during measurements could play a role [46,47,48].

A similar study performed in patients presenting with neurodegeneration appears to be a reasonable next step in this research. Such a survey would enable us to validate our initial concept in the context of neurological diseases. Importantly, a good explanation of the neurological benefits of sleeping on the side [12,13], as yet, has not yet been offered. In our previous paper, we hypothesized that in the lateral body position, because of decreased flow resistance in the extracranial veins, cerebral venous outflow is optimal, which in turn optimizes the activity of the glymphatic system. In this way, the brain is protected against neurodegeneration. In the present study on healthy individuals, this hypothesis was not confirmed. However, a different result in patients presenting with neurodegenerative disease cannot be ruled out. Therefore, such a study is needed.

Of note is that better functioning of the glymphatic system in the lateral body position has been demonstrated in animal studies [49]. It is known that the activity of the glymphatic system is primarily regulated by the noradrenergic system of the brain. Activation of the glymphatic system is dependent on a temporary decrease in cortical blood flow during the non-REM phase of sleep, which is followed by a macroscopic wave of inflow of cerebrospinal fluid. This change in cortical blood flow is regulated by noradrenergic neurons [21]. However, the reason noradrenergic stimulation should be increased while sleeping in the lateral decubitus position remains elusive. Perhaps noradrenergic neurons are stimulated by the vestibular nuclei that receive inputs from the semicircular canals, utricle, and saccule of the labyrinth. However, such anatomical connections have not yet been revealed (of note, a reverse stimulation from the locus ceruleus to the vestibular nuclei has been found [50,51]). Therefore, the hemodynamic hypothesis that an inflow of cerebrospinal fluid should be accompanied by an unrestricted outflow of venous blood seems encouraging.

Importantly, some modifications should be made to the design of future studies on this topic. We suspect that some of the counterintuitive changes in the sizes and flow volumes that were observed in our study resulted from compression of the IJV in its upper part by the transverse process of the atlas and/or the styloid process of the temporal bone [52]. To exclude such compression, the upper segment of the IJV, particularly the flow, should be carefully assessed in the slightly changed position of the head in relation to the neck. Abnormalities revealed through this test would require further diagnostic workouts, e.g., CT scans. It is known that flow abnormalities evoked by even minor stenoses of the IJV located in this area can be hemodynamically relevant [42]. In addition, such flow abnormalities can compromise the flow over the entire course of the IJV and possibly disturb the function of the jugular valve downstream [45].

In addition, in future studies on the importance of sleeping on the side, the pattern of cerebral venous outflow should be considered. Research on the flow in the IJVs has demonstrated that although in the majority of people, the IJVs constitute the main outflow route in the supine body position, the proportion of blood volume utilizing either the jugular or vertebral pathway can differ between individuals. Therefore, at least an approximate measurement of the flow in the vertebral veins should be performed in future studies.

In addition, the anatomy of the intracranial venous outflow should be taken into account. The main venous channels draining the brain comprise the superior sagittal sinus, inferior sagittal sinus, great cerebral vein, straight sinus, and occipital sinus. The superior sagittal sinus is the principal blood vessel draining the superficial parts of the cerebral hemispheres, particularly the cerebral cortex. Blood from the deep structures of the brain flows out through the deep venous system, which is composed of the internal cerebral veins and the basal veins of Rosenthal, and finally drains into the great cerebral vein, which in turn empties into the straight sinus. A perfectly symmetric pattern of the cerebral sinuses, with symmetric outflow to the left and right transverse sinuses from both the deep and superficial parts of the brain, is rarely encountered [35]. Typically, there is more or less asymmetric confluence of the sinuses, and in many people, there is a completely separated outflow to the right and left sides. Sometimes, the superficial structures of the brain are exclusively drained by the right transverse sinus, while the deep parts are drained by the left transverse sinus, or vice versa. Flow evaluation in the context of intracranial venous anatomy would require diagnostic imaging of these veins and sinuses. Such imaging could be performed low-invasively, preferentially using non-contrast methods, e.g., time-of-flight MR angiography [53]. In many neurological patients, such scanning would already be available for evaluation from the previous diagnostic workout. In others, it could be performed in addition to the scheduled control scans; thus, participation in the study would not add significant risks or inconveniences.

We acknowledge that our research has some limitations. The number of individuals assessed was rather small. In this relatively small group of participants, we found high heterogeneity in the outflow measurements. Although a higher number could theoretically facilitate drawing more valid conclusions, we suggest that a better understanding of cerebral venous outflow in the lateral body position should rather comprise an assessment of the intracranial outflow routes, as has been suggested above. As early as 2004, Doepp et al. found that although in the majority of humans in the supine body position, blood flows out of the brain mostly through the internal jugular veins, in about one-third of them, this outflow partially or even predominantly utilizes non-jugular pathways [54]. Most likely, similar outflow patterns, primarily associated with the anatomy of the main intracranial sinuses, affect venous circulation in the lateral decubitus body position. Therefore, there are several technical aspects of the measurements, as well as additional parameters that should perhaps also be evaluated, such as quantitative assessment of the flow in the extra-jugular veins of the neck. These issues should be considered in future studies on this topic. Nonetheless, the current study can be seen as a useful framework for future research on the role of cerebral and extracranial venous circulation in the settings of neurodegeneration.

## 5. Conclusions

In terms of venous outflow from the cranial cavity, the lateral decubitus body position did not differ significantly from the supine position. Our working hypothesis of a potentially protective effect against neurodegeneration of this body position during sleep through improved venous outflow from the brain has not been proven, at least in healthy persons. However, further studies on this topic, primarily in patients with neurodegeneration, are warranted.

## Figures and Tables

**Figure 1 jcm-14-01211-f001:**
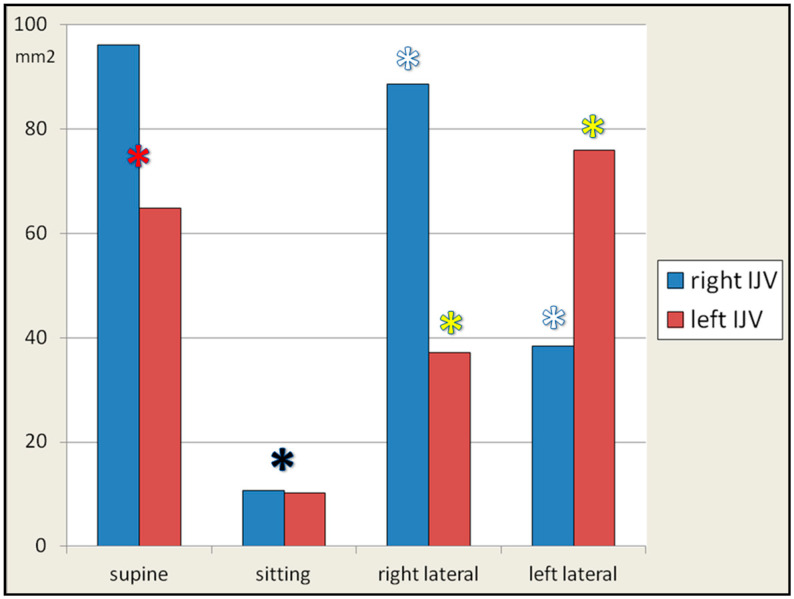
Cross-sectional areas of the right and left internal jugular veins in the supine, sitting, right, and left lateral decubitus body positions. Differences between the cross-sectional areas in particular body positions, assessed by means of the one-way ANOVA test, were statistically significant (*p* < 0.05). The differences between the cross-sectional areas in the supine, right lateral, and left lateral vs. sitting positions were statistically significant (paired *t*-test, *p* < 0.001). These significant differences were observed in both the right and left internal jugular veins (black asterisks). The cross-sectional area of each internal jugular vein significantly changed (white and yellow asterisks) during the switch from the right to the left lateral position, or vice versa (paired *t*-test: *p* < 0.001). The mean cross-sectional area of the right internal jugular vein was significantly (*p* = 0.03) larger than that of the left-sided vein (red asterisk).

**Figure 2 jcm-14-01211-f002:**
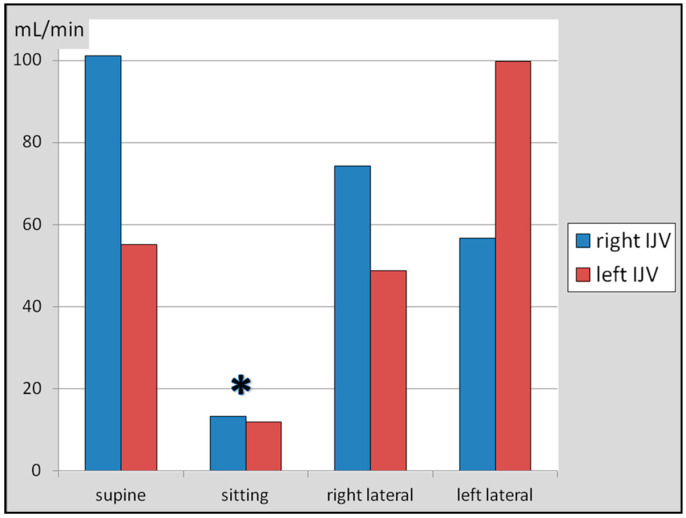
Flow volumes through the right and left internal jugular veins in the supine, sitting, right, and left lateral decubitus body positions. Differences between flow volumes in particular body positions, assessed by one-way ANOVA and paired *t*-test, were statistically significant (*p* < 0.05), but only for the sitting body position (asterisk). On the other hand, differences in the flow through the right or left IJV in the right and left lateral decubitus positions were not statistically significant (paired *t*-test: *p* > 0.05).

**Figure 3 jcm-14-01211-f003:**
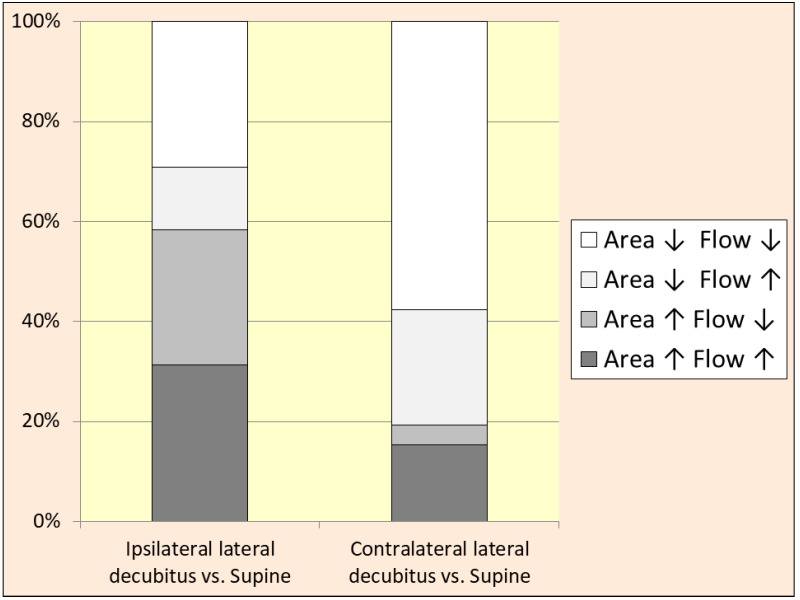
Change in the cross-sectional area and flow volume in the internal jugular veins in the lateral decubitus body position in comparison with the supine position; ipsilateral lateral decubitus—right IJV in the right lateral decubitus body position, or left IJV in the left decubitus body position; contralateral lateral decubitus—right IJV in the left lateral decubitus body position, or left IJV in the right decubitus body position; Area ↓: reduction of the cross-sectional area in comparison with the supine body position; Flow ↓: reduction of the flow in comparison with the supine body position; Area ↑: increase of the cross-sectional area; Flow ↑: increase of the flow.

**Figure 4 jcm-14-01211-f004:**
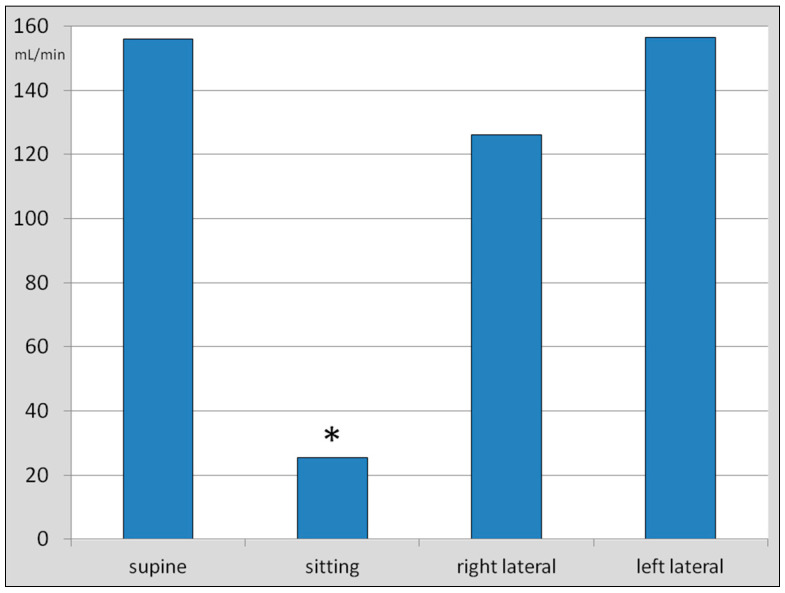
Total flow volumes in both internal jugular veins in the supine, sitting, right, and left lateral decubitus body positions. Flow volume in the sitting body position was significantly different from other positions (asterisk), as demonstrated by the one-way ANOVA test and the paired *t*-test. Differences between flow volumes in other body positions were not statistically significant (*p* > 0.05).

**Figure 5 jcm-14-01211-f005:**
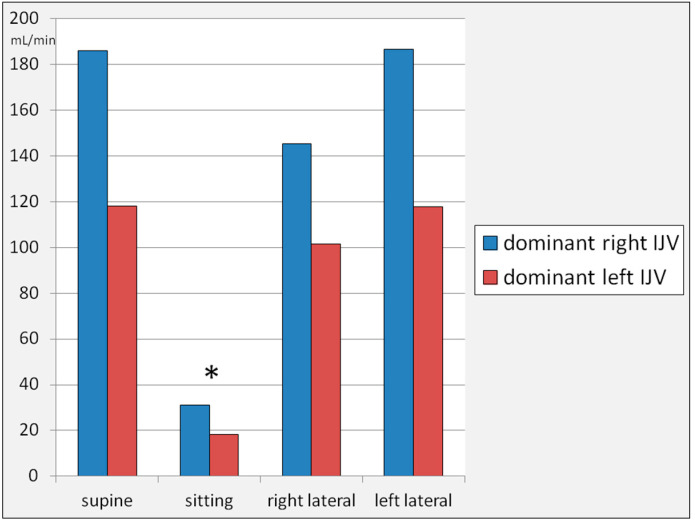
Total flow volumes in both internal jugular veins in the supine, sitting, right, and left lateral decubitus body positions in individuals presenting with dominant right- or left-sided outflow. Flow volume in the sitting body position was significantly different from other positions (asterisk), as has been demonstrated by the one-way ANOVA test and the paired *t*-test. The differences between the flow volumes in the other body positions were not statistically significant (*p* > 0.05). In addition, there were no statistically significant differences between total flow volumes in the individuals presenting with right vs. left IJV dominance regarding flow volumes in the supine, right lateral, and left lateral positions.

## Data Availability

Anonymized data presented in this study are available upon request from the corresponding author. The preprint version of this paper is available at https://www.preprints.org/manuscript/202408.2265/v1 (Posted: 03 September 2024).

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
