# Peer review of "Blood Flow in the Internal Jugular Veins in the Lateral Decubitus Body Position in the Healthy People"

_jcm, 2025, doi:10.3390/jcm14041211_

Round 1
Reviewer 1 Report
Comments and Suggestions for Authors
Thanks for the opportunity to review your paper. This interesting study looked at cross-sectional areas and flow volumes in the internal jugular veins during lateral decubitus positioning.
Some comments for your consideration:
- Suggestions for improving readability/flow of the abstract - Starting off with "Optimal venous..." instead of "Of as yet..."; replacing the phrase "Of as yet" with a more common phrase perhaps "to date"?; adding demographics of the healthy volunteers; adding values of cross-sectional areas etc. in results
- Main text some revisions for language - line 68 ("The" current study?); line 70 (typo error "in in"); line 73 (remove "the" and keep just "healthy subjects"?) etc.
- Methods may be useful to add more details on how the positions were standardised and maintained throughout the measurements
- Results may be useful to report mean difference so that this can be discussed in terms of clinical significance too; also the data does not seem to be normally distributed - should a nonparametric be used instead of t-test?
- Should the lack of control of head rotation be recognised as a limitation also?
Author Response
We express gratitude for pointing out the limitations of our paper.
- We rearranged and supplemented the abstract, as has been suggested:
“Background. Some studies suggested that the lateral decubitus position during sleep may protect the brain from neurodegenerative processes. Although mechanisms of such a possible protection is not known, an optimal venous outflow may be responsible. Venous outflow from the cranial cavity is dependent on the body position. Yet, to date, flow in the internal jugular veins (IJVs) in the lateral position has not been studied quantitatively. Methods. Using ultrasonography, we measured the cross-sectional areas and flow volumes in the IJVs in a group of 25 healthy individuals, aged 20-52 ± 12.1 years. These measurements were performed in the supine, upright and lateral decubitus positions. Results. In the lateral decubitus positions we revealed a collapse of the IJV located higher, and dilatation of the opposite vein, and a shift of flow from one vein to the opposite one. In the right lateral position the mean cross-sectional area and flow in the right IJV were: 88.6 ± 71.1 mm2 and 74.3 ± 97.5 mL/min, in the left IJV: 37.2 ± 33.4 mm2 and 48.8 ± 82.8 mL/min. In the left lateral position ‒ in the right IJV: 38.4 ± 30.7 mm2 and 56.7 ± 56.1 mL/min, in the left IJV: 75.9 ± 51.9 mm2 and 99.7 ± 123.9 mL/min. Yet there was also a high heterogeneity of the cross-sectional area changes and in many participants this pattern was not observed. Regarding flow volumes in the lateral body positions, in comparison with the supine one the total outflow through both internal jugular veins was not significantly different. Conclusions. In terms of venous outflow, the lateral decubitus position does not differ significantly from the supine one. Working hypothesis of a potentially protective effect of this body position during sleep against neurodegeneration, through improved venous outflow, has not been proven, at least in the healthy persons.”
- We corrected errors, as has been requested.
- “Methods may be useful to add more details on how the positions were standardised and maintained throughout the measurements ".
We added more information on the methodology of measurements performed:
“In all of them, ultrasonographic examinations of the IJVs in the middle portion of these veins (at the level of the thyroid cartilage) were performed. All the measurements were done at the same level, which was marked on the skin with a white crayon. These examinations were done on a flat examination table, and during measurements the participants were asked to breathe normally. Measurements of the cross-sectional areas and flow volumes were performed on both sides of the neck. The cross-sectional areas were calculated in a standard way, after freezing the picture of the target vein and using the measuring application of the ultrasound system. The mean velocity flow was evaluated during one respiratory cycle and the flow volume was calculated using the values of the cross-sectional area and mean velocity flow. During the scanning, participants were asked not to rotate their heads, also in the lateral body position. All the above-described measurements were performed in four body positions…:”
- “Results may be useful to report mean difference so that this can be discussed in terms of clinical significance too; also the data does not seem to be normally distributed - should a nonparametric be used instead of t-test?”
We added information on the assessment of the normality of data in the Statistical analysis chapter:
“Normality of data was assessed using the Shapiro-Wilk test, which revealed that the distribution of the samples was not significantly different from a normal distribution”
- “Should the lack of control of head rotation be recognised as a limitation also”
We added explanation in the Methods chapter:
“During the scanning, participants were asked not to rotate their heads, also in the lateral body position”
Reviewer 2 Report
Comments and Suggestions for Authors
The article by Kawalec-Rutkowska et. al., titled "Blood flow in the internal jugular veins in the lateral decubitus body position in healthy people" investigates blood flow dynamics in the internal jugular veins (IJVs) of 25 healthy individuals across different body positions—supine, sitting, and lateral decubitus—using ultrasonography.
It explores whether the lateral decubitus position optimizes venous outflow and could potentially protect against neurodegeneration through enhanced glymphatic system function. Results showed that while IJVs exhibited positional changes, such as vein collapse and compensatory dilation, total venous outflow in lateral decubitus was not significantly different from the supine position.
The study concludes that this body posture does not improve venous outflow in healthy individuals, though further research is recommended in populations with neurodegenerative conditions. Limitations include a small sample size, exclusion of alternative venous pathways, and high heterogeneity in flow patterns.
The overall quality of the article should be improved, and the following changes are needed.
· The hypothesis in the introduction assumes an association between venous outflow and neurodegeneration based on indirect evidence. More emphasis on how the study directly addresses this gap could enhance clarity.
· The sample size (n=25) is relatively small and thus it limits the study’s generalizability.
· There are missing details on participant recruitment and demographic variability (e.g., weight, height), which may influence IJV dynamics.
· Lack of blinding in ultrasonographic assessments may introduce observer bias.
· The heterogeneity in cross-sectional area and flow volume changes in the lateral position is acknowledged but not fully analyzed. Are there patterns based on anatomy or demographics?
· The lack of significant differences between supine and lateral positions for total flow volume contrasts with findings from prior studies, yet the authors do not explore this discrepancy rigorously.
· The authors claims’ about the protective role of lateral sleeping positions in neurodegeneration remain speculative without direct evidence from patients with these diseases.
Figure 1 and 2 lacks statistical significance on the graphs. Please indicate them on the graphs and mention the exact p value in the legends.
Author Response
Thank you for reviewing the manuscript.
- “The hypothesis in the introduction assumes an association between venous outflow and neurodegeneration based on indirect evidence. More emphasis on how the study directly addresses this gap could enhance clarity..”
We added more detailed explanation on a possible role for venous incompetence as one of the cause of glymphatic system impairment, leading to neurodegeneration:
“Importantly, cerebral veins, as well as extracranial venous outflow routes, including the internal jugular veins, are closely linked to the glymphatic system function, since they provide key drainage pathways for metabolic waste removal from the brain parenchyma. During activation of the glymphatic sytem, the cerebrospinal fluid enters the parenchyma through the periarterial spaces. This process is facilitated by aquaporin-4 water channels located on the astrocytes. Then, the cerebrospinal fluid mixes with the interstitial fluid, and this process enables removing the above-mentioned toxic proteins. Since the cerebrospinal fluid‒interstitial fluid mixture moves along the perivenous spaces, any disruption to the venous outflow may potentially contribute to the glymphatic dysfunction and to increased risk of the neurodegenerative disease”
- “The sample size (n=25) is relatively small and thus it limits the study’s generalizability”
We agree that the sample size was quite small, and we have already acknowledge this fact in the Discussion. However, our results and also previous research suggest that in order to get better and reliable results, more complex evaluation of the outflow instead of increasing the sample size should be done. At the moment we are underway at planning such a next step in this research.
We added a commentary on these issues:
“We acknowledge that there are some limitations of our research. The number of assessed individuals was rather small. In this relatively small group of participants, we found high heterogeneity of the outflow measurements. Although a higher number could theoretically facilitate drawing more valid conclusions, we suggest that a better understanding of the cerebral venous outflow in the lateral body position should rather comprise assessment of the intracranial outflow routes, as has been above suggested. As early as in 2004, Doepp et al. found that although in the majority of humans in the supine body position blood flows out of the brain mostly through the internal jugular veins, in about one third of them this outflow partially or even predominantly utilizes the non-jugular pathway. Most likely, similarly different outflow patterns, primarily associated with anatomy of main intracranial sinuses, regard venous circulation in the lateral decubitus body position. Therefore, there are several technical aspects of the measurements, as well as additional parameters that perhaps should also be evaluated, such as quantitative assessment of the flow in the extra-jugular veins of the neck. These issues should be considered during future studies on this topic.”
- “There are missing details on participant recruitment and demographic variability (e.g., weight, height), which may influence IJV dynamics.”
We added information of the method of the recruitment for this study. Since the internal jugular veins primarily drain the brain (and partially soft tissues of the face), and cerebral flow is not dependent on such anthropometric variables and the height of body weight, but rather on the size of brain and activity of this organ. We included information that these participants were not obese, since obesity may indirectly influence cerebral circulation:
“For this purpose, we recruited 25 healthy individuals, 21 women and 4 men. They were recruited from the employees of our university. They were aged 20-52 years, mean 33.5 years. None of them presented with significant co-morbidities, including obesity.”
- “Lack of blinding in ultrasonographic assessments may introduce observer bias”
Actually, since this research was performed on healthy individuals only, their clinical status (no neurological disease) could not be blinded to the observer, since there was only one variable. Other viable, such as body position during examination for obvious reasons (ultrasonographic examination) could not be blinded.
- “The heterogeneity in cross-sectional area and flow volume changes in the lateral position is acknowledged but not fully analyzed. Are there patterns based on anatomy or demographics?”
Results of analysis aimed at explanation of this heterogeneity have been already provided:
“Since there was a high heterogeneity regarding the cross-sectional areas and flow volumes in the examined veins, we also analyzed if an asymmetry of IJVs had any association with the parameters studied. It is well known that the IJVs exhibit high anatomical variability. In most of people these veins are asymmetric. The right IJV is usually wider, although in some people the left IJV is dominant. Also, the inflow to these veins is typically asymmetric. A symmetric confluence of the sinuses (the connection of the superior sagittal sinus, the straight sinus, and the occipital sinus, which divides into two transverse sinuses) is rarely seen. In a majority of people this anatomical structure is asymmetric. Hence the inflow into the IJVs is not equal and usually the cerebral outflow is primarily directed to the right IJV.
We categorized our participants into two groups: with dominant right IJV, and those with dominant left IJV. For the purpose of this analysis, the dominance of a particular IJV was defined as the vein, in which the flow in the supine body position was higher than in the opposite one. In the studied group there were 14 participants with dominant right IJV and 11 participants with dominant left IJV. Flow volumes in the dominant-right-IJV vs. dominant-left-IJV individuals are shown in Figure 5. In general, flow volumes were higher in the group with dominant right IJV, although these differences were not statistically significant. There was a statistically significant flow reduction through the IJVs in the sitting position, while the differences regarding flow volumes between other body positions were statistically insignificant.”
However, this analysis was rather inconclusive. Consequently, we provided a framework for future studies in this area that could explain paradoxical behavior of some IJV examined in this study:
“…. not only we revealed a high heterogeneity of the cross-sectional areas changes, which was manifested through very high standard deviations of this parameter, but also we found that in the lateral decubitus body position in many participants the IJVs did not follow the above-described pattern (see: Figure 3). It suggests that the position-dependant change of the shape and flow in the IJV is more complex and is determined by more factors than gravitation alone”
and:
“Also, anatomy of the intracranial venous outflow should be taken into account. Main venous channels draining the brain comprise the superior sagittal sinus, the inferior sagittal sinus, the great cerebral vein, the straight sinus and the occipital sinus. The superior sagittal sinus is the principal blood vessel draining the superficial parts of the cerebral hemispheres, particularly the cerebral cortex. Blood from the deep structures of the brain flows out through the deep venous system, which is composed of the internal cerebral veins and the basal veins of Rosenthal, which finally drain into the great cerebral vein, which in turn empties into the straight sinus. A perfectly symmetric pattern of the cerebral sinuses, with symmetric outflow to the left and right transverse sinuses from both deep and superficial parts of the brain, in rarely encountered [33]. Typically, there is a more or less asymmetric confluence of the sinuses, and in many people there is a completely separated outflow to the right and left side. Sometimes, the superficial structures of the brain are exclusively drained by the right transverse sinus, while deep parts by the left one, or vice versa. Flow evaluation in the context of the intracranial venous anatomy would require a diagnostic imaging of these veins and sinuses. Such an imaging could be done low-invasively, preferentially using the non-contrast methods, e.g. time-of-flight MR angiography”
- “The lack of significant differences between supine and lateral positions for total flow volume contrasts with findings from prior studies, yet the authors do not explore this discrepancy rigorously”
Actually, the flow in the internal jugular veins in the lateral body position was not studied before. This study, to the best of our knowledge, is the first one. Our previous pilot study (Simka, M.; Czaja, J.; Kowalczyk, D. Collapsibility of the internal jugular veins in the lateral decubitus body position: a potential role of the cerebral venous outflow against neurodegeneration. Med Hypothes 2019, 133, 109397) evaluated a few individuals and looked at cross-sectional diameters of these veins only.
There is already an explanation regarding this issue in the text:
“In our previous pilot study [17], we found that in the lateral decubitus position the IJV, which in this body posture was above the level of the right atrium, was partially collapsed, while the opposite one remained open and was slightly wider in comparison with the cross-sectional area of this vein in the supine body position. When the patient changed body position to the opposite lateral one, for example from the left to the right side, the IJVs behaved following this pattern: the vein located above the heart level collapsed, and that located below dilatated. Still, in this preliminary study the IJV were not examined quantitatively and only a few individuals were assessed. The current study was aimed at quantitative evaluation of the flow and diameters in the IJVs in different body positions, particularly in the lateral decubitus ones, in order to better understand flow phenomena in these veins occurring when people are lying down on their side. “
- “The authors claims’ about the protective role of lateral sleeping positions in neurodegeneration remain speculative without direct evidence from patients with these diseases.”
Actually, there is the study by Levendovsky (Levendowski et al., Head position during sleep: potential implications for patients with neurodegenerative disease. J Alzheimer Dis 2019, 67,631-638.) where a higher prevalence of neurodegenerative disease has been found in patients who slept longer in the supine vs. lateral position. There are also two other studies supporting this hypothesis:
Gnarra, O.; Calvello, C.; Schirinzi, T.; Beozzo, F.; De Masi, C.; Spanetta, M.; Fernandes, M.; Grillo, P.; Cerroni, R.; Pierantozzi, M.; et al. Exploring the association linking head position and sleep architecture to motor impairment in parkinson’s disease: An exploratory study. J Pers Med 2023, 13, 1591. https://doi.org/10.3390/jpm13111591
Girolami, S.; Tardio, M.; Loredana, S.; Di Mattia, N.; Micheletti, P.; Di Napoli, M. Sleep body position correlates with cognitive performance in middle-old obstructive sleep apnea subjects. Sleep Med X 2022, 4, 100050.
Thus, there is an evidence, although quite slim. A protective role of the lateral position during sleep is in line with research conducted in rodents, where a better function of the glymphatic system during sleep in the lateral position was revealed.
All these studies have already been citied in our paper.
“Of note, some studies suggested that body position during sleep may have an impact on the development of neurodegeneration [12-14].”
- We supplemented the legends of the Figures 1 and 2, as has been suggested, and added asterisks indicating statistical significances.
We also added some amendments in the Results chapter:
“These differences were statistically significant (the paired t-test: P = 0.03). This finding is not new and it is well known that the right IJV is usually wider, which is associated with embryological development of these veins”.
- We also added information on anatomical connections between the locus ceruleus nucleus and the vestibular nuclei, with citations:
“(of note, a reverse stimulation, from the locus ceruleus to the vestibular nuclei have been found.”
Round 2
Reviewer 1 Report
Comments and Suggestions for Authors
Thanks for making the suggested changes. The methods are more clearly described now and the abstract reads better. I would still suggest reporting of mean difference if this is possible.
Author Response
We included statistics for differences in means (amended parts of the manuscript are underlined ). Sorry for misunderstanding of your message.
Reviewer 2 Report
Comments and Suggestions for Authors
The authors have satisfactorily answered all my queries
Author Response
Thak you for your valuable comments and acceptance of the revised manuscript